

# Can the growth of deltaic shorelines be unstable?

Meng Zhao[1], Gerard Salter[2], Vaughan R. Voller[3], and Shuwang Li[4]

[1]Department of Mathematics, University of California, Irvine, CA 92697, USA
[2]Department of Earth Sciences, University of Minnesota, Minneapolis, MN 55455, USA
[3]Department of Civil, Environmental, and Geo- Engineering, St. Anthony Falls Laboratory, 500 Pillsbury Drive SE, Minneapolis, MN 55455, USA
[4]Department of Applied Mathematics, Illinois Institute of Technology, Chicago, IL 60616, USA

**Correspondence:** Vaughan R. Voller (volle001@umn.edu)

**Abstract.** We study a sedimentary delta prograding over a fixed adversely sloping bathymetry, asking whether a perturbation to the advancing shoreline will grow (unstable) or decay (stable) through time. To start, we use a geometric model to identify the condition for acceleration of the shoreline advance (autoacceleration). We then model the growth of a delta on to a fixed adverse bathymetry, solving for the speed of the shoreline as a function of the water depth, foreset repose angle, fluvial topset slope, and shoreline curvature. Through a linearization of this model, we arrive at a stability criterion for a delta shoreline; indicating that autoacceleration is a required for an unstable growth. This is the first time such a shoreline instability has been identified and analyzed. We use the derived stability criterion to identify a characteristic lateral length-scale for the shoreline morphology resulting from an unstable growth. On considering example experimental and field conditions we observe that this length scale is typically larger than other geomorphic features in the system, e.g., channel spacings and dimensions, suggesting that the signal of the shoreline growth instability in the landscape might be "shredded" by other surface building processes, e.g., channel avulsions and along shore transport.

## 1 Introduction

Shorelines are the moving boundary between land and sea, and their evolution is of great importance to the estimated ten percent of the global population that live in their proximity (Wong et al., 2014). Shorelines are also an area of scientific interest, because their shape records information about the processes that formed them. While significant progress has been made in characterizing shoreline shape (Shaw et al., 2008; Geleynse et al., 2012), inferring formative processes from shoreline shape remains a challenge. Galloway (1975) recognized that qualitatively, the shape of a delta shoreline reflects the relative importance of waves, tides, and fluvial input, but using shoreline shape to assess the strength of these processes quantitatively remains an open challenge (Nienhuis et al., 2015; Baumgardner, 2016). Part of the challenge may lie in the susceptibility of shorelines to instabilities. For example, an instability associated with high-angle waves results in the self-organization of regular, quasiperiodic shoreline features (Ashton and Murray, 2006). Another type of instability important for deltaic shorelines is the channel-forming instability. Although unchannelized sheet flow can be observed in nature on some alluvial fans, channelized flow is more common. This has been ascribed to the instability of sheet flow, tending to evolve towards a channelized state



Whipple et al. (1998). This instability can be expected to manifest itself in the shape of the shoreline, with areas near channels receiving the most sediment and therefore prograding faster relative to the rest of the shoreline.

Here our interest will focus on a new mechanism that might drive the instability of an advancing delta shoreline. Our motivation is the recent works from Hajek et al. (2014) and López et al. (2014), who have studied the growth of a sedimentary delta under a condition of a "back-tilted" subsidence rate; a condition that resulted in the water depth ahead of the shoreline decreasing with distance (i.e., the delta builds on an adverse slope). Such scenarios can arise in foreland basins where the sediment supply is sufficiently high relative to subsidence for progradation to occur, if a prograding delta approaches the opposite side of a lake or reservoir, or if the delta toe encounters an adverse slope on an offshore bar. In a one-dimensional modeling and experimental study López et al. (2014) indicated that, for some combinations of sediment input and subsidence style, delta progradation on an adverse slope could exhibit a positive acceleration; referred to as *autoacceleration*. We think that such a behavior could be a critical ingredient for the onset of unstable growth. To see this, imagine a two-dimensional, in plan view, growth scenario with an advancing planar shoreline front. Under an autoaccelerating regime any "blip" (perturbation) in the growth direction along the shoreline front could find itself in a location which is more favorable for growth. In this way, it is possible that, under the right conditions, rather than being consumed by the advancing planar shoreline, this "blip" will accelerate away and provide a potential driver for an unstable morphological break down of the planar shoreline. Indeed, the two-dimensional delta growth experiments from Hajek et al. (2014) underscore this possibility by observing "a tendency for shorelines to *run away* seaward in response to base-level fall in back-tilted basins".

In exploring the possible instability associated with autoacceleration, we will appeal to the analogy between solid/liquid phase change processes and delta shoreline advance (Swenson et al., 2000; Voller et al., 2004; Capart et al., 2007; Lorenzo-Trueba et al., 2009; Voller, 2010; Ke and Capart, 2015; Lai et al., 2017). This analogy is based on the construction of a shoreline mass balance condition, equating the sediment flux arriving to the rate of its advance—a condition directly analogous to the phase change interface heat balance Stefan condition in melting problems ( Crank (1984)). The original shore balance proposed by Swenson et al. (2000) has been recently modified by (Ke and Capart, 2015) to account for the shoreline planform curvature. Recognizing the extensive work related to the role of curvature in the morphological instability of growing interfaces (Mullins and Sekerka, 1963; Sekerka et al., 2015; Paterson, 1981; Li et al., 2004, 2009; Zhao et al., 2016), this modification allows us to expand the so called Swenson/Stefan analogy to develop a criterion for an unstable delta shoreline advance.

Principally, we are interested in answering a number of key questions:

- Under what conditions would an unstable shoreline growth arise and how would it evolve over time?

- What, if any, is the connection between autoacceleration and an unstable shoreline growth?

- What would be the characteristic length scale of the instability and how does this scale compare to other geomorphic length scales in deltaic shoreline settings, e.g., channel spacings?

To set the stage for our study, we adopt the delta geometry used in the López et al. (2014) model and then, on invoking the additional simplifying assumption of a static basin with a constant water level and fixed floor bathymetry, we arrive at



an explicit criterion for the onset of autoacceleration. To see and understand how such a condition may lead to an unstable growth condition, we further perform a linear stability analysis of the Ke and Capart (2015) shoreline condition, identifying the criterion when a specified small perturbation on a planar autoaccelerating shoreline front would be expected to grow, i.e., become unstable.

## 2  A Geometric Model

The one-dimensional model recently present in López et al. (2014) assumed that the growth of a delta into a basin with a back-tilting hinged subsidence rate would, under the supply of a unit sediment flux $q$ at the origin $x = 0$, maintain a similar geometry with constant positive topset ($S_T > 0$) and foreset ($S_F = \tan(\alpha)$) slopes. Here we retain these geometric assumptions but invoke an additional assumption that the delta builds onto a basement with a fixed (non-subsiding) slope $S_B$; a limiting simplification, that allows us to directly arrive at an explicit condition under which autoacceleration will occur. This geometric model is schematically represented in the cross-section (long profile) shown in Fig.1. If we assume that this schematic is for a one-dimensional planar growing delta, an analysis of the change in area of the deposit cross section due to a small incremental advance of the shoreline $x = \ell(t)$, leads to the following expression for the shoreline speed

$$v = \frac{d\ell}{dt} = \frac{q}{\ell S_T + D}, \tag{1}$$

where $D = L\sin(\alpha)$ is the water depth at the point where the foreset toe meets the basement, $\alpha = \in [0, \frac{\pi}{2}]$ is the angle of the foreset, $L(\ell)$ is the length of the foreset slope, and $q$ is the unit flux of sediment plus pore space. On taking a further derivative in time we arrive at an expression for the acceleration of the shoreline

$$a = \frac{d^2\ell}{dt^2} = -q\frac{d\ell}{dt}\frac{[S_T + L'(\ell)\sin(\alpha)]}{[\ell S_T + L\sin(\alpha)]^2}, \tag{2}$$

where, $L'(\ell)$ is the rate of change of the foreset length $L(\ell)$ with shoreline position, which, on noting that $dD/d\ell = S_B$, can be written as $L'(\ell) = S_B/\sin\alpha$ . To exhibit autoacceleration, the value of $a$ will need to be positive, requiring that the numerator in the last term on the right hand side of Eq.(2) will need to be negative, which, in turn, implies that,  under the assumption of a fixed basement, an explicit condition for autoacceleration can be written as

$$S_B < -S_T. \tag{3}$$

This simply states that autoacceleration would be expected when the basement slope at the toe of the foreset is *adverse* (negative) with a value that exceeds the value of the topslope, i.e., $|S_B| > S_T$. As we noted above, while exceeding this condition might lead to unstable shoreline growth it is not clear if the occurrence of autoacceleration is a necessary condition for such a behavior. For example, the geometry (e.g., curvature) of a shoreline perturbation on an accelerating front might retard its further growth. In order to arrive at a more rigorous condition for shoreline stability, we need to develop a treatment that can account for plan-form perturbations of the planar front. Such a treatment will require a more sophisticated model for the partitioning of the sediment between the fluvial and submarine. Towards this end, we develop a linear stability analysis for a two-dimensional plan view shoreline that uses the local shoreline mass balance proposed by Ke and Capart (2015).



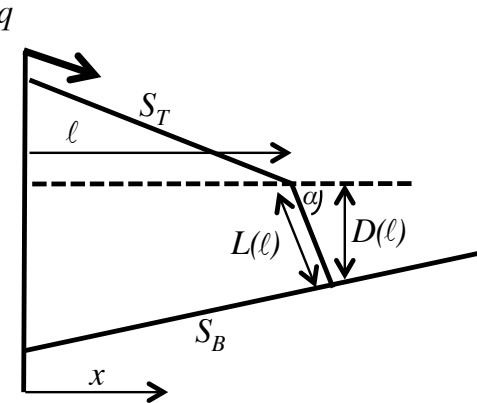

**Figure 1.** Schematics of sediment delta cross-sections depositing on to a fixed basement with an adverse slope $S_B < 0$. The topset slope is $S_T > 0$ and the submarine foreset has angle $\alpha$, length $L(\ell)$, and a depth of $D(\ell)$ at the point where its toe touches the basement.

## 3 A Linear Stability Analysis

The key ingredient in the Swenson analogy Swenson et al. (2000) between the advance of a sediment delta front into a standing body water and the tracking of the liquid/solid Stefan melting front is the determination on how the sediment arriving on the land side of the shoreline is deposited into the submarine. In the one-dimensional Swenson analogy this involves a simple

distribution of the excess sediment arriving at the shoreline to maintain a submarine foreset of constant slope, see Fig.1, a device that leads to a relationship between the speed of the shoreline advance and the land-side sediment supply. The major contribution in the work by Ke and Capart (2015) is to generalize this relationship to a case where the growing delta has a two-dimensional planform ($x$ in the seaward direction and $y$ in the lateral), i.e., from Eq. (23) in Ke and Capart (2015), the shoreline evolves as

$$\frac{\partial \mathbf{x}}{\partial t} \cdot \mathbf{n} = \frac{\mathbf{J} \cdot \mathbf{n}}{\sin\alpha(L(\mathbf{x}) + \frac{1}{2}\kappa L^2(\mathbf{x})\cos\alpha)}, \tag{4}$$

where $\mathbf{x}$ is the Cartesian position vector for a point on the shoreline, $\mathbf{J} \cdot \mathbf{n}$ is the unit sediment flux (+ pore space) arriving to the landward side of the shoreline (essentially the excess material that can be used for shoreline advance), $\mathbf{n}$ is the seaward pointing unit normal on the shoreline, $\alpha$ is the angle of repose of the foreset, $L(\mathbf{x})$ is its length, and $\kappa$ is the plan-form curvature of the shoreline. We will use this more general shoreline condition as the basis for our linear stability analysis.

In the case of a planar shoreline (curvature $\kappa = 0$) at position $x = \ell(t)$, under our geometric assumption of a fixed a fluvial

slope, $\mathbf{J} \cdot \mathbf{n} = q - S_T \ell \dot{\ell}$ and the condition in Eq. (4) reduces to

$$\dot{\ell} = \frac{q - S_T \ell \dot{\ell}}{L \sin\alpha} \tag{5}$$

where $\dot{\ell} = d\ell/dt$. On recognizing that $L\sin\alpha = D$—the depth at the foreset toe—we see that this is simply a rearrangement of our geometric model in Eq.(1). The starting point for our stability analysis is to introduce a small perturbation of the planar



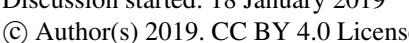


front with the form

$$x(t,y) = \ell(t) + \epsilon\delta(t)\cos(ky), \tag{6}$$

where, with reference to Fig. 2, $\delta(t)$ is the amplitude of the perturbation, the parameter $\epsilon \ll 1$, and $k$ is the wave number, related to the wavelength of the perturbation through $\lambda = 2\pi/k$. This step allows us to ask whether a small perturbation to the shoreline will shrink back to the advancing front (stable) or if it will accelerate away from it (unstable)? With the given

perturbation, we note that, to first order $O(\epsilon)$, the velocity vector of the front and the shoreline sediment flux vector at any given lateral location $y$, are still in the $x$-direction, i.e.,

$$\frac{\partial \mathbf{x}}{\partial t} \cdot \mathbf{n} = \dot{\ell} + \epsilon\dot{\delta}\cos(ky), \tag{7}$$

and

$$\mathbf{J} \cdot \mathbf{n} = q - S_T \ell\dot{\ell} - \epsilon S_T (\ell\dot{\delta} + \dot{\ell}\delta)\cos(ky). \tag{8}$$

In addition we note that curvature of the perturbation is given by

$$\kappa = \epsilon k^2 \delta\cos(ky), \tag{9}$$

and the foreset length at any given lateral position $y$ is

$$L(y) = L(\ell) + \epsilon L'(\ell)\delta\cos(ky) = L(\ell) + \epsilon\frac{S_B(\ell)}{\sin(\alpha)}\delta\cos(ky), \tag{10}$$

On substitution of these expansions Eqs.(7—10) into the shoreline condition Eq.(4), after some algebra and the matching of $O(1)$ and $O(\epsilon)$ terms, we arrive at the following relationships for the rate of shoreline advance and perturbation amplitude growth

$$\dot{\ell} = \frac{q}{\sin\alpha L(\ell) + S_T\ell}, \tag{11}$$

$$\dot{\delta} = -\left[\frac{\sin\alpha\left(\frac{S_B}{\sin(\alpha)} + \frac{L^2(\ell)}{2}k^2\cos\alpha\right) + S_T}{L(\ell)\sin\alpha + S_T\ell}\right]\dot{\ell}\delta. \tag{12}$$

Thus, under the assumptions employed to this point, the shoreline stability rests on the amplitude rate in Eq.(12) taking on

a value $\dot{\delta} < 0$. On noting the strictly nonnegative nature of most of the terms in this expression, it follows that for an unstable growth—an increase of the perturbation amplitude with time—the numerator

$$I = \underbrace{S_B(\ell)}_{\text{destabilizing term}} + \underbrace{\frac{k^2 L^2(\ell)}{2}\cos\alpha\sin\alpha + S_T}_{\text{stabilizing term}}. \tag{13}$$





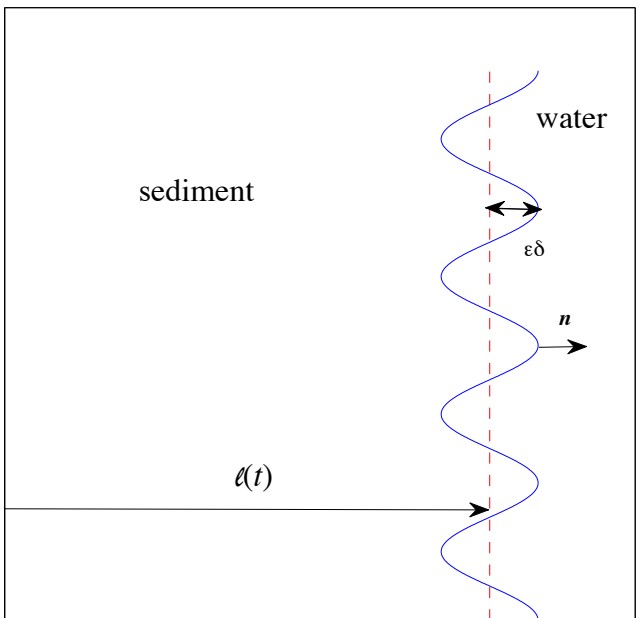

**Figure 2.** Schematic diagram of a perturbed shoreline

needs to be negative, $I < 0$. From this, recalling that the foreset slope $S_F = \tan(\alpha)$ and that the depth at its toe is $D = L\sin(\alpha)$, we arrive at the following condition for unstable shoreline growth

$$S_B^e < -\frac{k^2 D^2}{2 S_F}, \tag{14}$$

where we have defined the effective basement slope as $S_B^e = S_B + S_T$. This criterion states that an unstable growth requires the presence of an adverse effective basement slope $S_B^e = S_B + S_T < 0$, i.e, unstable shoreline growth requires the autoacceleration condition $S_B < -S_T$. Indeed, we note that in the limit of $\alpha \to \pi/2$, where the foreset slope becomes a "cliff face", the stability criterion is identical to the autoacceleration condition.

At this point we need to emphasis two possible limitations of our analysis. In the first place while Ke and Capart (2015) offers the most general and correct treatment available for the relationship between sediment supply and shoreline front advance it is limited by the assumptions of a constant water level and fixed basement bathymetry. Secondly our treatment neglects the possible role of lateral sediment transport (Ikeda, 1982; Parker, 1984). Hence, a strict interpretation of any findings based on our stability criterion needs to carry the rider that they may only be applicable to systems where subsidence, sea-level changes, and the role of lateral sediment transport can be ignored . Never the less, we feel that the consequences of this condition, examined in detail below, reveal critical features on the nature of delta shoreline growth.





## 4   Disscussion

Now that we have established that the condition of auto-acceleration can lead to unstable growth of a Delta shoreline we need to consider two issues. How, under a given set of conditions, will a shoreline instability evolve? and What length scales (wave-lengths) will the resulting instability exhibit?

### 5   4.1   Evolution of the Instability

In our analysis of the instability the obvious place to start is to explore the shape of the stability region and develop an understanding of how unstable shoreline perturbations might evolve with time. To provide a physical context that enables us to analyze our stability criterion under conditions that are consistent with realizable experimental systems we consider the XES10 experiment reported in Hajek et al. (2014), an experiment specifically designed to study the growth of shoreline in the presence

of a back-tilted (adverse) subsidence.

To illustrate the shoreline stability region, under XES10 conditions, we use Eq.(14) to plot the water depth $D(\ell)$ against the basement slope $S_B(\ell)$ for four different values of the foreset slope $S_F = \tan(\alpha)$, Fig. 3[a]. In these plots we have set the wave number $k = 1$ and, consistent with XES10, we have set the topset slope as $S_T = 0.03$ (Hajek et al., 2014). It is evident that the unstable region gets larger as $S_F$ increases. In particular, the most unstable scenario (corresponding to $\alpha = \pi/2$)

is, as noted above, the criterion for autoacceleration. To further explore these stability plots , let us consider three points $P_A, P_B$ and $P_C$ with $S_B = -0.2$ and $S_F = 1$. The point $P_A(S_B = -0.2, D = 0.94)$ belongs to the stable region indicating the shoreline perturbation decays, $\dot{\delta}(t) < 0$. The point $P_B(S_B = -0.2, D = 0.583)$ is exactly on the boundary separating the stable and unstable regions, indicating the growth rate of the perturbation is zero, $\dot{\delta}(t) = 0$. The $P_C(S_B = -0.2, D = 0.2)$ is in the unstable region indicating the shoreline perturbation grows, i.e. $\dot{\delta}(t) > 0$.

In our study of evolution of an unstable shoreline, we will neglect subsidence and assume that the final basement profile, (reported in Fig. 2 in Hajek et al. (2014)) is fixed thorough time. In this way, between 1.6 and 5 meters from the sediment source we will have an effective adverse basement slope of $S_B^e = S_B - S_T = -0.2 + 0.03 = -0.17$ and, on assuming a foreset slope of $S_F = \tan(\pi/4) = 1$, a linear water depth at the forest toe of $D(x) = 0.95 - 0.2(x - 1.6)$ m $1.6 \leq x \leq 5$. In this context, we will consider the advance of a shoreline on this basement with the slightly perturbed initial shape $x(0) = \ell(0) + \delta(0)\cos(y)$,

where $\ell(0) = 1.65$ m, $\delta(0) = 0.05$ m, and lateral extent of $y \in [0, 2\pi]$ m. With these values, on scaling the time so that the input unit flux is $q = 1$, the analytical solution of the linear theory in Eq. (12) gives

$$\delta(\ell) = 0.528 \frac{e^{\ell(-0.615 + 0.0588\ell)}}{(7.47 - \ell)^{0.852}} \text{ m} \tag{15}$$

where the advance of the bulk shoreline with time is

$$\ell(t) = \frac{1.27 - \sqrt{0.979 - 0.37t}}{0.17} \text{ m} \tag{16}$$

In Fig. 3[b], we plot the absolute size of the perturbation $\delta$ as a function of the bulk shoreline position $\ell$. The shoreline starts from the stable point $P_A$ with $D = 0.94$. In its initial progradation is in a stable regime and the amplitude of the perturbation



decreases. The minimum amplitude 0.0389 is reached at $\ell = 3.43$, point $P_B$. Here the the growth rate of the perturbation is zero but beyond this point we enter the unstable regime where the perturbation grows and the shoreline becomes unstable (e.g see $P_C$).

We can also use the above conditions to test the the validity of the linear theory used in the derivation of the stability criterion, Eq. (14). In particular, following an approach used in previous works (Li and Li, 2011; Zhao et al., 2016) we have developed a semi-implicit boundary element like scheme to compute the nonlinear dynamics of a shoreline. In these nonlinear computations, we measure the growth of the perturbation as $\delta(t) = \max ||\mathbf{x}| - \ell(t)|$, where $\mathbf{x}$ is the position vector of the shoreline. The linear prediction are seen to be in excellent agreement with our nonlinear results, see Fig. 3[b], in particular we note that, in the non-linear analysis, the minimum perturbation 0.0391 is reached at position $\ell = 3.429$—values close to the linear analysis counterparts of 0.0389 and 3.43. Moreover, we have performed a series of simulations using different initial perturbations, and confirmed that the difference between the linear and nonlinear results is indeed $O(\epsilon^2)$.

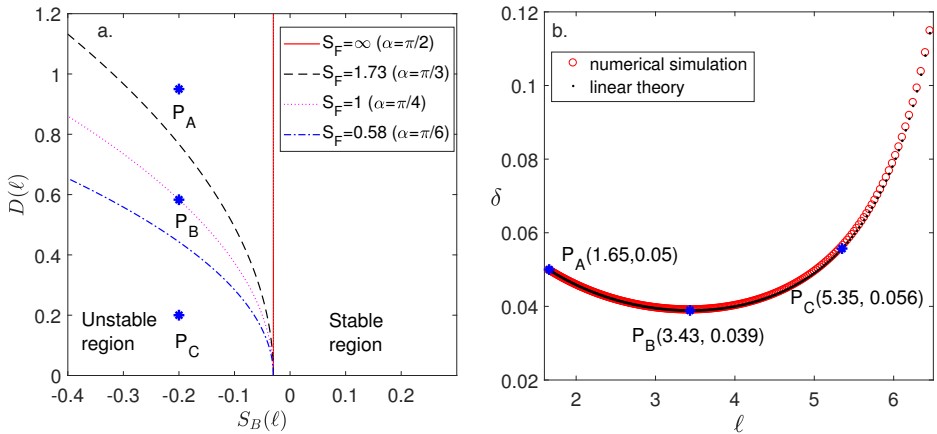

**Figure 3.** [a] The stability region of the absolute criterion for different foreset slopes $S_F$ with $k = 1$ and $S_T = 0.03$. When $S_F = 1$, the point $P_A$ is in a stable region, the point $P_B$ is on a boundary between the stable and unstable region, and point $P_C$ is in an unstable region. [b] The amplitude of the shape perturbation $\delta$ as a function of bulk shoreline position $\ell$, in the case where the initial shape of the shoreline $x(t = 0) = 1.65 + 0.05 \cos(y)$, $S_F = 1$, and a linear the depth, $D(x) = 0.95 - 0.2(x - 1.6)$, at the foreset toe.

## 4.2 Choice of characteristic Length scale

Following the typical approach of a morphological instability analysis (see Sekerka et al. (2015) ) we can look for two characteristic wavelengths associated with our shoreline perturbations. The first of these is the wavelength associated with the fastest growing wave number; given sufficient time, we would expect this to be the dominant wavelength of the evolving instability. The second, is the the wavelength associated with the wave number at which the amplitude of the perturbation neither grows or decays—the neutral wavelength.



In the case that the initial perturbation exhibits a number of modes, $x(y,t) = \ell(t) + \epsilon \sum_{k=1}^{\infty} \delta_k(t) \cos(ky)$, each mode independently evolves following Eq. (12). In this circumstance, we can determine, essentially by direct inspection of Eq. (12), that the fastest growing wave length would be associated with the wave number $k = 0$, corresponding to an infinitely long wave length—recall that wavelength $\lambda = \frac{2\pi}{k}$. This presents something of a conundrum, while an infinite wavelength is math-

5 ematically consistent with our analysis it is unlikely to be physically achievable. Rather, we would expect that the dominant wave-length observed, in a given system, would be set by the lateral size of the system ( e.g., the width of an experiment, or the distance between channels).

Perhaps a better length scale to characterize the nature of unstable shoreline growth is the neutral wavelength. On appropriate rearrangement, this wavelength can be calculated by substitution of the wave number definition $k = \frac{2\pi}{\lambda_n}$ in to our the stability

criterion (eq.(14)),

$$\lambda_n = \frac{\sqrt{2}\pi D}{\sqrt{-S_F S_B^e]}}, \quad -S_B > 0, \tag{17}$$

The value of $\lambda_n$ provides us with a minimum lateral length-scale for the resulting morphology of the growth of an unstable shoreline.

### 4.3 Values of Neutral Wavelength in Experimental and Field Systems)

Our contention is that, determining the possible values of the neutral wave length in experimental and field systems will inform

15 us on the expected length-scales of the instability in delta shoreline growth along adverse basement slopes.

As an example, let us again consider the limit conditions found in the XES10 Hajek et al. (2014) experiment ($S_B = -0.2, S_T = 0.03, S_F = 1$), with an adverse slope, defined by the water depth $D = 0.95 - 0.2(x - 1.6)$. In this case, as the shoreline advances (autoaccelerates $S_B > S_T$) onto the adverse slope, the neutral wavelength Eq.(17) linearly decreases from a value of $\lambda_n(1.6) = 10.24$m to a value, at the maximum length of the experiment, of $\lambda_n(5) = 2.91$m. Hence, in this exper-

20 imental system, we see the influence of the unstable growth on the dynamics of the shoreline motion is at a large-scale, at or beyond the lateral dimension of the system, $y = 3$m (Hajek et al., 2014). Note, extending the length of the adverse slope to the point where the water depth $D \to 0$, would allow smaller wavelengths to become unstable. For example, at $x = 6.3$m ($D = 0.1$m), the neutrally stable wavelength is $\lambda_n(6.3) = 0.11$m. At this point, however, there is a very limited remaining longitudinal domain over which the instability can develop.

As a field example, we consider the Wax-lake/Atchafalaya Bay area in the Gulf of Mexico. Due to diversions in the Mississippi river system this has been a site of active delta building over the last forty years, Wagner et al. (2017). We also have access to 1935 pre-growth bathymetry data (Atchafalaya Bay, https://www.ngdc.noaa.gov/mgg/bathymetry/estuarine/). We stress that our intention here is not to model the growth of specific deltas in this system but rather to use the pre-delta bathymetry data to provide constraints on the spacial extent and values of pre-existing basement slope regimes in a field setting. Toward this

objective, we have selected a sampling region (3km lateral, 6km longitudinal),10km off-shore of the Atchafalaya outlet. Figure 4 shows the location of three longitudinal profiles that span this system. These profiles indicate that, on average, the sample region has a persistent adverse slope $S_B = 0.00015$ in the longitudinal $x$ (off-shore) direction, along which the water depth





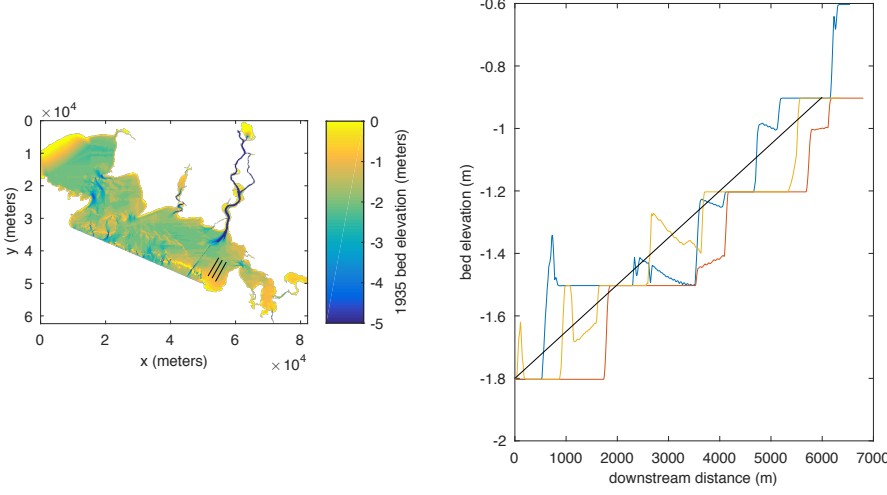

**Figure 4.** Atchafalaya 1935 bathymetry data. The left panel shows the location of 3 profiles of length ∼6km, the profiles start ∼ 10 km off-shore, are a direction normal to the shoreline, and cover a lateral range of 3km. The panel on right provides the bed elevations along each of the profiles; the average slope of these profiles, is taken as $S_B = 0.00015$

changes according to $D = 1.8 - 0.00015(x - 10000)$. If we assume that the foreset is $S_F = 1$ and set the topset slope as $S_T = 0.000071$ (a value consistent with the current day fluvial slope on Wax Lake delta (Wagner et al., 2017) we see that, as a shoreline advances along this adverse slope, the predicted neutral wavelengths Eq.(17) are on the order of $\sim 1km$, linearly increasing with off-shore distance $x$, from a value of $\lambda_n(x = 10km) \approx 900$m to a value of $\lambda_n(x = 16km) \approx 450$m.

## 5 Conclusions

In this work we have used a geometric model and a linear stability analysis to investigate conditions under which the progradation of a planar sedimentary delta shoreline over a non-subsiding basement could become unstable, i.e., a condition where a perturbation of the shoreline will grow faster than the shoreline advance. In this work we find that:

– A geometric model provides a simple condition for autoacceleration, stating that when a delta is building on to a fixed adverse slope, it is expected that the shoreline will exhibit a positive seaward acceleration if the value of the basement slope is larger than the topset (fluvial) slope.

– A linear stability analysis shows that, in an autoacceleration condition, the growth of a delta shoreline prograding on a fixed adverse slope will become unstable, i.e., lateral perturbations on the shoreline, greater than a particular neutral wavelength, will grow faster than its bulk advance.

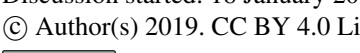


- The analysis indicates that the fastest (dominant) growth perturbation wavelengths, are at the lateral size of the system under consideration.

- In experiment and field systems the neutral wavelength of the perturbations (the wavelength at which there is no growth or decay) is also relatively large; in excess of the widths of experimental systems and beyond typical distributary channel spacings in the field .

Thus while we have clearly provided a positive answer to the question of this paper, "Can the growth of a deltaic shoreline be unstable?" we have also shown that in most experimental and field delta systems it may not be possible to distinguish the signal of an unstable growth. In other words while delta building along an adverse basement slope is unstable the resulting signal of the shoreline growth instability in the landscape will probably be "shredded" by other surface building processes, e.g., channel avulsions and along shore transport.

*Competing interests.* The authors have no competing interest

*Acknowledgements.* SL would like to thank the support from National Science Foundation through grants DMS-1720420 for partial support. SL also acknowledge the partial support through grant ECCS-1307625. GS acknowledges funding from the National Science Foundation Graduate Research Fellowship under Grant No. 00039202. The authors are also grateful for insightful discussion with Professor C. Paola and for the insightful and helpful comments from Andrew Ashton. The data supporting the conclusions of this work are self contained in the mathematical analysis presented.



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
