# Peer review of "Can the growth of deltaic shorelines be unstable?"

_Earth Surface Dynamics, 2018_

## Referee Comment (RC1) · John Shaw (Referee) · 13 Feb 2019

Review of Zhao et al. "Can Growth of Deltaic Shorelines be Unstable" By John Shaw, University of Arkansas (shaw84@uark.edu)

This study considers the possibility of a deltaic shoreline (or topset-foreset rollover) that grows unstably over an adversely sloping basement. Using a geometric modeling approach and linear stability analysis, it is found that for certain conditions, a linearly advancing delta shoreline will have perturbations grow. It is also found that the most unstable wavelength is infinitely long, meaning that any unstable growth will likely be the size of the container. In it also interesting that once a delta enters the unstable region, perturbations will keep growing more unstable for the simple geometries discussed here.

While intentionally abstract, this mathematical approach allows for the growth of instabilities to be investigated in a very pure and clear manner. I have found the mathematics to be sound. The paper is also very well written and the figures are sufficient.

The main point of improvement is that the applications chosen to illustrate the theory are somewhat cursory compared to the theory. I find the application to Atchafalaya Bay in particular to be too simplified. The 6 km long transects showing gradual shallowing are very focused in a small part of the bay, and might not be characteristic of the slopes that a delta progrades over. Then, the prediction of the stable wavelength is given for x = 10 and 16 km, which is far longer than the adverse bedslope measurements. Also, the S_F is roughly -0.00024 for the Wax Lake Delta, as reported by Shaw et al. (2016) and cannot be reasonably estimated as S_F=-1. This would increase the neutral wavelength and instability as described in Eq. 17 and 14.

Ultimately, I would consider trying to find more or better examples of deltas prograding across adversely sloping beds. Leva Lopez et al. (2014) provide a good discussion that might yield another geological case study. Deltas forming near or underneath glaciers are a potentially great place to look (Carlson et al., 1999; Dowdeswell and Vásquez, 2013; Lønne and Nemec, 2011. . . these are not perfect but show potential). This effort might really broaden the appeal of this paper beyond theoreticians (like me).

P4L16: I initially thought that this equation was incorrect because l_dot was on both sides of the equation. I am now sure that it is correct, but it may be good to show this equation solved for l_dot.

P7L13: I do not understand how a wavenumber k = 1 is chosen from the XES10 conditons.

P9L11: shouldn't S_B always be positive? This looks like S_B must be negative.

Works Cited in this Review Carlson, P. R., Cowan, E. A., Powell, R. D. and Cai, J.: Growth of a post-Little Ice Age submarine fan, Glacier Bay, Alaska, Geo-Mar. Lett., 19(4), 227–236, doi:10.1007/s003670050113, 1999. Dowdeswell, J. A. and Vásquez,

M.: Submarine landforms in the fjords of southern Chile: implications for glacimarine processes and sedimentation in a mild glacier-influenced environment, Quat. Sci. Rev., 64, 1–19, doi:10.1016/j.quascirev.2012.12.003, 2013. Leva Lopez, J., Kim, W. and Steel, R. J.: Autoacceleration of clinoform progradation in foreland basins: theory and experiments, Basin Res., 26(4), 489–504, doi:10.1111/bre.12048, 2014. Lønne, I. and Nemec, W.: Modes of sediment delivery to the grounding line of a fast-flowing tidewater glacier: implications for ice-margin conditions and glacier dynamics, Geol. Soc. Lond. Spec. Publ., 354(1), 33–56, doi:10.1144/SP354.3, 2011. Shaw, J. B., Mohrig, D. and Wagner, R. W.: Flow patterns and morphology of a prograding river delta, J. Geophys. Res. Earth Surf., 2015JF003570, doi:10.1002/2015JF003570, 2016.

---

## Author Comment (AC1) · 11 Mar 2019

We wish to thank Reviewer #1 (John Shaw) for his insightful and helpful comments. In the interest of promoting discussion, we have addressed this review in advance of submitting our revision. The specific points raised by the reviewer are addressed below:

The main point of improvement is that the applications chosen to illustrate the theory are somewhat cursory compared to the theory. I find the application to Atchafalaya Bay in particular to be too simplified. The 6 km long transects showing gradual shallowing are very focused in a small part of the bay, and might not be characteristic of the slopes that a delta progrades over.

Our purpose is not to model the growth of the Atchafalaya Bay deltas per se. We agree that the adverse slopes we measured are not characteristic of the slopes that the deltas prograde over – the pre-delta bathymetry is fairly uniform in the areas where the deltas are growing. Rather, the purpose of this section is to obtain realistic values of our model parameters based on a field setting, and thereby constrain the wavelengths that we could realistically expect to find in the field.

Then, the prediction of the stable wavelength is given for x = 10 and 16 km, which is far longer than the adverse bedslope measurements. Also, the S_F is roughly -0.00024 for the Wax Lake Delta, as reported by Shaw et al. (2016) and cannot be reasonably estimated as S_F=-1. This would increase the neutral wavelength and instability as described in Eq. 17 and 14.

We have removed the reference to the x coordinate, since our model is independent of the definition of the x coordinate. However, we nevertheless do find that the neutral wavelength is larger than the distance over which we measure the adverse slope. This reinforces our conclusion that the shoreline instability would probably not be observed, at least for this system. We have corrected our value of S_F, and find that this indeed increases the value of the neutral wavelength. We have also added a sentence to emphasize our conclusion that under the parameters we obtained from this field setting, stable shoreline growth is predicted. Our text (P10L2) will be modified as follows:

> …we see that, as a shoreline advances along this adverse slope, the predicted neutral wavelengths (Eq.17) are, compared to the system size, relatively large. Linearly decreasing with off-shore distance, we obtain values ranging from $\lambda_n \approx$ 63 km to $\lambda_n \approx$ 32 km. The two deltas growing in the modern Atchafalya Bay are around 10 km in diameter, smaller than the predicted neutral wavelength, so we are led to conclude that if these advancing deltas were to encounter an adverse slope, the indication of the resulting unstable growth would not be easily observable.

Ultimately, I would consider trying to find more or better examples of deltas prograding across adversely sloping beds. Leva Lopez et al. (2014) provide a good discussion that might yield another geological case study. Deltas forming near or underneath glaciers are a potentially great place to look (Carlson et al., 1999; Dowdeswell and Vásquez,

2013; Lønne and Nemec, 2011. . . these are not perfect but show potential). This effort might really broaden the appeal of this paper beyond theoreticians (like me).

We agree that adverse basement slopes should be relatively common in proglacial deltas, for example if the delta reaches a moraine, the wall of a fjord, or perhaps progradation reaching the flexural bulge. However, we were not able to find any clearly documented examples of proglacial deltas from which we could estimate an adverse bed slope. Ultimately, we chose Houseknecht et al. (2001), which was also cited by Lopez et al. (2014), as an additional example. We will add the following paragraph following our Atchafalaya Bay example:

> As a larger-scale field example, we consider the Torok formation in the Colville basin, as reported by Houseknecht et al. (2001). This formation displays clinoforms prograding over an adverse basement slope associated with a foredeep. Based on the schematic cross-section shown in Figure 7B of Houseknecht et al. (2001), we can estimate the adverse basement slope over which the shelf-margin prograded. Over a distance of roughly 200 km, we measure a steady decrease in the clinoform height from around 1900 to 710 m. Assuming that the clinoform heights correspond to basin depth, a minimum estimate of $|S_B|$ is $6*10^{-3}$ . This estimate is a minimum, because it does not account for relative sea level rise, which would cause the basin depth to increase over time. We measure a foreset slope of roughly 0.03, which is consistent with typical values for continental slopes. While we do not have an estimate for $S_T$ available, it is reasonable to assume that it is small relative to the basement slope we measured. Based on these values, we obtain an estimate for the neutral wavelength $\lambda_n$ that ranges from 629 to 237 km, decreasing as the shelf margin progrades into shallower water. The cross-section reported in Houseknecht et al. (2001) spans a distance of 450 km, so we see that the estimated neutral wavelengths are on the order of the system size.

In light of our modified calculations of the neutral wavelength, we will also alter the conclusion as follows:

> -In experiment and field systems the neutral wavelength of the perturbations (the wavelength at which there is no growth or decay) is expected to be large, in excess of the widths of experimental systems, and well beyond delimiting field length scale such as distributary channel spacings.

> Thus while we have clearly provided a positive answer to the question of this paper, "Can the growth of a deltaic shoreline be unstable?" we can also conclude that observing clear signals of unstable growth in typical experimental and field delta systems would be unlikely. In other words, while delta building along an adverse basement slope is unstable, the resulting signal of the shoreline growth instability in the landscape will probably be "shredded" by other surface building processes, e.g., channel avulsions and along-shore transport.

P4L16: I initially thought that this equation was incorrect because I_dot was on both sides of the equation. I am now sure that it is correct, but it may be good to show this equation solved for I_dot.

We will add this in our revision.

P7L13: I do not understand how a wavenumber k = 1 is chosen from the XES10 conditons.

The stated purpose of section 4.1 is to illustrate the nature of the evolution of the stability region. To put this in context we have chosen to do this using the XES data. The choice of wave number is somewhat arbitrary in such illustrative calculations and here, for convenience we have chosen k=1. We will modify the text as follows:

> In making these plots, for convenience of presentation, with no real loss of generality, we have arbitrarily set the wave number k = 1. Further, to establish consistency with XES10, we have set the topset slope as $S_T$ = 0.03 (Hajek et al., 2014).

P9L11: shouldn't S_B always be positive? This looks like S_B must be negative.

Our convention is that S_B should be negative for an adverse basement slope. We will add the following text after equation (3).

> Recall the basement slope $S_B$ is defined as change of water depth in the direction of the delta growth, dD/dl = $S_B$. Thus Eq.(3), simply states that autoacceleration would be expected when the delta is growing into shallowing water, such that the basement slope at the toe of the foreset is adverse (negative) with a value that exceeds the value of the topslope, i.e., $|S_B| > S_T$.

We also found a few instances in the text where we incorrectly reported the sign of S_B. This will be corrected in our revision.

---

## Referee Comment (RC2) · Jorge Lorenzo-Trueba (Referee) · 24 Mar 2019

Zhao et al. explore the possibility of unstable shoreline growth on fluvial deltas under adverse basement slopes. The authors first present a cross-shore geometric model that describes and quantifies how an acceleration in the shoreline migration rate (previously observed in flume experiments) can occur. The authors then extend the model to account for small oscillations in the plant-view shoreline geometry to study their effect on shoreline advance. Using perturbation analysis, the authors find the range of basement slopes, water depths, and length scales in which unstable growth can occur. After discussing potential connections with previously reported flume experiments and field observations from the Wax-lake delta, the authors conclude that shoreline instabilities due to shallowing depths, although present, most likely cannot be separated from other environmental signals.

This publication is an important contribution as it is the first one (to my knowledge) to explore and quantify the magnitude and occurrence of unstable shoreline growth due to shallowing ocean depths. I include a few comments that aim at helping improve the manuscript. After these comments are addressed, I recommend the paper to be accepted for publication.

My main question is regarding the relationship presented in line 18, page 3 (i.e., dL/dl = S_B/sin(alpha)). When alpha=90 degrees, the equation provides a relationship I believe to be correct. However, when tan(alpha)=S_B I believe the symmetry in the geometry should result in dL/dl = S_B/2. Additionally, when alpha«< , this equation suggests that dL/dl»>. I am not sure I understand why this is the case. My guess would have been that when alpha«< a change in "l" would result in a small change in "L".

My derivation results in dL/dl = 1/(1/tan(alpha) + 1/S_B). I might be wrong, but this solution seems to get the right answer in the scenarios presented above.

Although I believe this equation would not change the overall results significantly, it would affect equations (10), (12), and (13), which are part of the perturbation analysis. Thus, the equations that describe the criteria for unstable shoreline progradation would also change.

Page1: Line 6: . . . autoacceleration is required for unstable to occur. . .

Page 4: I suggest the authors clarify in Figure 1 the sign of the basement slope S_B, which is negative in this case. I would do the same for the topset slope S_T.

Page 6: Line 7: . . . to emphasize. . . Line 12: Nevertheless

Page 8: Line 8: . . . linear prediction is in excellent agreement. . .

Hope these comments help Jorge Lorenzo-Trueba
* * *
**ESurfD**
[Figure]

2019.

---

## Author Comment (AC2) · 19 Apr 2019

**Revised and Combined Reply to Reviewers**

April 18, 2019

In this document, we list the reviewers' comments in black, and our response in blue. Line numbers correspond to lines in the manuscript with highlighted changes, where P#L# denotes the page and line numbers.

**Response to Reviewer #1**

We wish to thank Reviewer #1 (John Shaw) for his insightful and helpful comments. This response is a slight modification of our previous reply. The specific points raised by the reviewer are addressed below:

The main point of improvement is that the applications chosen to illustrate the theory are somewhat cursory compared to the theory. I find the application to Atchafalaya Bay in particular to be too simplified. The 6 km long transects showing gradual shallowing are very focused in a small part of the bay, and might not be characteristic of the slopes that a delta progrades over.

Our purpose is not to model the growth of the Atchafalaya Bay deltas per se. We agree that the adverse slopes we measured are not characteristic of the slopes that the deltas prograde over – the pre-delta bathymetry is fairly uniform in the areas where the deltas are growing. Rather, the purpose of this section is to obtain realistic values of our model parameters based on a field setting, and thereby constrain the wavelengths that we could realistically expect to find in the field.

Then, the prediction of the stable wavelength is given for $x = 10$ and 16 km, which is far longer than the adverse bedslope measurements. Also, the $S_F$ is roughly $-0.00024$ for the Wax Lake Delta, as reported by Shaw et al. (2016) and cannot be reasonably estimated as $S_F = -1$. This would increase the neutral wavelength and instability as described in Eq. 17 and 14.

Note in revised text Eq. 14 is now 13 and Eq. 17 is now 16. We have reworked our paragraph on Wax Lake Delta, starting at P9L30. We have removed the reference to the $x$ coordinate, since our model is independent of the definition of the $x$ coordinate. Our finding that the neutral wavelength is larger than

the distance over which we measure the adverse slope, reinforces our conclusion that the shoreline instability would probably not be observed, at least for this system. We have corrected our value of $S_F$, and find that this indeed increases the value of the neutral wavelength. We have also added a sentence to emphasize our conclusion that, under the parameters we obtained from this field setting, unstable shoreline growth would not be observable (p10L12).

Ultimately, I would consider trying to find more or better examples of deltas prograding across adversely sloping beds. Leva Lopez et al. (2014) provide a good discussion that might yield another geological case study. Deltas forming near or underneath glaciers are a potentially great place to look (Carlson et al., 1999; Dowdeswell and Vásquez, 2013; Lønne and Nemec, 2011. . . these are not perfect but show potential). This effort might really broaden the appeal of this paper beyond theoreticians (like me).

We agree that adverse basement slopes should be relatively common in proglacial deltas, for example if the delta reaches a moraine, the wall of a fjord, or perhaps progradation reaching the flexural bulge. However, we were not able to find any clearly documented examples of proglacial deltas from which we could estimate an adverse bed slope. Ultimately, we chose Houseknecht et al. (2001), which was also cited by Lopez et al. (2014), as an additional example. We have added a new paragraph based on this example starting at P10L14.

Additionally, based on our new calculations of the neutral wavelength, we have modified our conclusions (P11L7, P11L10) to emphasize that the wavelength of the predicted instability is large.

P4L16: I initially thought that this equation was incorrect because $\dot{\ell}$ was on both sides of the equation. I am now sure that it is correct, but it may be good to show this equation solved for $\dot{\ell}$.

Done (P5L5).

P7L13: I do not understand how a wavenumber k = 1 is chosen from the XES10 conditons.

The stated purpose of section 4.1 is to illustrate the nature of the evolution of the stability region. To put this in context we have chosen to do this using the XES data. The choice of wave number is somewhat arbitrary in such illustrative calculations and here, for convenience we have chosen $k = 1$. We have added this clarification to P7L19.

P9L11: shouldn't $S_B$ always be positive? This looks like $S_B$ must be negative.

Our convention is that $S_B$ should be negative for an adverse basement slope. We now clarify this at P3L22. We also found multiple instances in the text where we incorrectly stated the sign of $S_B$, and these have been corrected.

**Response to Reviewer # 2**

We thank Reviewer #2 (Jorge Lorenzo-Trueba) for providing helpful comments and for spotting the error in our geometric model.

My main question is regarding the relationship presented in line 18, page 3 (i.e., $dL/d\ell = S_B/sin(\alpha)$). When $\alpha = 90$ degrees, the equation provides a relationship I believe to be correct. However, when $tan(\alpha) = S_B$ I believe the symmetry in the geometry should result in $dL/d\ell = S_B/2$. Additionally, when $\alpha \ll$ , this equation suggests that $dL/dl \gg$. I am not sure I understand why this is the case. My guess would have been that when $\alpha \ll$ a change in $\ell$ would result in a small change in $L$. My derivation results in $dL/d\ell = 1/(1/tan(\alpha)+1/S_B)$. I might be wrong, but this solution seems to get the right answer in the scenarios presented above. Although I believe this equation would not change the overall results significantly, it would affect equations (10), (12), and (13), which are part of the perturbation analysis. Thus, the equations that describe the criteria for unstable shoreline progradation would also change.

We are very grateful to JLT for spotting this error. We have modified our geometric model, now correctly associating the water depth with the delta toe, not the shoreline. We have now incorporated this modification into our analysis and adjusted the calculations throughout the paper as appropriate. JLT is correct that this adjustment does not change the main result or finding of the paper; essentially, as we note in the revised paper, it simply modifies the definition of the effective basement slope used in our geometric and stability treatments. We note, however, that this modification is more pleasing, from a physical point of view, since it now, in addition to the topset and basement slopes, includes a dependence on the foreset slope into the analysis. Our correction is given at P3L15, and carried into equation 2 and P3L17. We then introduce an "effective" basement slope in equation 3, which allows the rest of the analysis to remain the same as before, after substituting in the effective basement slope in place of $S_B$. This correction slightly changes the stability criterion, as seen in the new equation 13 (equivalent to eq. 14 in original submission) , as well as the neutral wavelength (equation 16—eq.17 in original submission). We have updated the calculations throughout the paper in light of this correction, and we find that the neutral wavelengths are slightly larger than before.

Page1: Line 6: . . . autoacceleration is required for unstable to occur. . . In revised text this line now reads—autoacceleration is a necessary condition for unstable growth.

Page 4: I suggest the authors clarify in Figure 1 the sign of the basement slope $S_B$, which is negative in this case. I would do the same for the topset slope $S_T$. This is similar to the issues raided by John Shaw— as noted in the reply to that review-we have taken care to ensure that the signs of the slopes are now well defined. In particular we note slope signs in both the graphic and caption of Figure 1.

Page 6: Line 7: . . . to emphasize. . . Line 12: Nevertheless Correction made

Page 8: Line 8: . . . linear prediction is in excellent agreement. . . Correction made

**Author changes**

In addition to the corrections in response to reviewers, in order to streamline the revised manuscript we have also removed eq 13 and the text around it from the original manuscript